# A *Pseudomonas* Lysogenic Bacteriophage Crossing the Antarctic and Arctic, Representing a New Genus of *Autographiviridae*

**DOI:** 10.3390/ijms24087662

**Published:** 2023-04-21

**Authors:** Zhenyu Liu, Wenhui Jiang, Cholsong Kim, Xiaoya Peng, Cong Fan, Yingliang Wu, Zhixiong Xie, Fang Peng

**Affiliations:** College of Life Sciences, Wuhan University, Wuhan 430072, China

**Keywords:** *Pseudomonas*, prophage, polar regions, comparative genomic, novel viral cluster, *Autographiviridae*

## Abstract

Polar regions tend to support simple food webs, which are vulnerable to phage-induced gene transfer or microbial death. To further investigate phage-host interactions in polar regions and the potential linkage of phage communities between the two poles, we induced the release of a lysogenic phage, vB_PaeM-G11, from *Pseudomonas* sp. D3 isolated from the Antarctic, which formed clear phage plaques on the lawn of *Pseudomonas* sp. G11 isolated from the Arctic. From permafrost metagenomic data of the Arctic tundra, we found the genome with high-similarity to that of vB_PaeM-G11, demonstrating that vB_PaeM-G11 may have a distribution in both the Antarctic and Arctic. Phylogenetic analysis indicated that vB_PaeM-G11 is homologous to five uncultured viruses, and that they may represent a new genus in the *Autographiviridae* family, named *Fildesvirus* here. vB_PaeM-G11 was stable in a temperature range (4–40 °C) and pH (4–11), with latent and rise periods of about 40 and 10 min, respectively. This study is the first isolation and characterization study of a *Pseudomonas* phage distributed in both the Antarctic and Arctic, identifying its lysogenic host and lysis host, and thus provides essential information for further understanding the interaction between polar phages and their hosts and the ecological functions of phages in polar regions.

## 1. Introduction

Polar environments contain diverse communities of bacteria, archaea, and fungi, which, together with viruses, form an essential part of the polar ecosystem [1]. Microorganisms present in polar regions are often exposed to extreme conditions, such as extreme cold, freeze–thaw cycles, high radiation, desiccation, high salinity, and low nutrient status, so they must be highly metabolically versatile and exceptionally tolerant [2]. The differences between the microbial communities at the two poles are significant; for example, 78% of the operational taxonomic units (OTUs) in the surface layer of Southern Ocean seawater and 70% of the OTUs in the surface layer of Arctic Ocean seawater are unique [3]. Moreover, even the identical bacterial OTUs with identical 16S rRNA genes from opposite poles have some degree of genomic diversity [4].

*Pseudomonas* is one of the most ecologically important genera of bacteria [5]. Some *Pseudomonas* species have the ability to utilize toxic and harmful compounds, leading to various applications in environmental biotechnology and bioremediation [6]. Some *Pseudomonas* species have produced bioactive compounds that can be used as food preservatives or novel antimicrobial agents in the food or pharmaceutical industries [7]. Moreover, some *Pseudomonas* species are pathogenic, such as the multi-drug resistant (MDR) *Pseudomonas aeruginosa*, an opportunistic pathogen that is the leading cause of nosocomial infection with high lethality [8]. Some extremely cryophilic *Pseudomonas* species are also the source of cryophilic enzymes suitable for industrial applications [9]. With their enormous metabolic diversity and their extraordinary ability to tolerate various environmental conditions leading to their ubiquitous ecological distribution, *Pseudomonas* spp. are dominant among the culturable bacteria in the Arctic and Antarctic [6].

Viruses are the most enriching microorganisms on earth, and it is estimated that approximately 20% of the microbes present in the ocean are killed by viruses daily [10,11]. Phages are believed to play a vital role in marine ecosystems through lysic host bacteria to control microbial community dynamics, host metabolic status, and biogeochemical cycles [1]. Upon infection of host cells, temperate phages can usually integrate their genomes into bacterial chromosomes. When the lytic cycle is initiated, they may be excised along with the DNA of the host bacterium and transferred to a new bacterial host by transduction [12]. Temperate phages are common in many sequenced bacterial genomes, and both the prophage and its lytic cycle can affect individual bacteria or populations in different ecosystems [13,14]. Lysogenic phages can protect host bacteria from further infection, provide host virulence factors, help hosts form biofilms, tolerate antibiotics, and grow under oligotrophic environments, and some lysogenic phages can act as regulatory switches for host bacteria [15].

In polar regions, where microorganisms are at the core of the simple food webs, phages play a more critical role in these extremely cold habitats [16]. However, knowledge of polar phage diversity and the role phages play in microbial physiology and ecology is still limited due to the difficulty of obtaining environmental samples from polar regions and the presence of large numbers of non-culturable phage hosts. Current developments in metagenomic technologies have greatly assisted in the discovery and understanding of unknown viruses [17]. An analysis of metagenomic data indicated that a large number of phages are in the polar areas and are widespread in various environments, including the ocean [18], sea ice [19], soil [20,21], lakes [22], and cryoconite holes [23]. However, most of the virus sequences in metagenomics datasets cannot be attributed to known viruses, and such viruses are known as viral dark matter [24]. The vast majority of phages are known only from sequence fragments, and understanding of their ecological functions and biological characteristics is greatly hampered by the lack of studies on their host range and gene function [19,25]. Furthermore, current attempts to identify putative bacterial hosts based on metagenome sequence homology are limited by viral sequence variation and must be complemented by phage–bacteria system identification [26]. However, only a few phage–bacteria systems have been isolated from polar regions. To our knowledge, only four virus–host systems have been isolated from Arctic sea ice and four culturable viruses have been isolated from Antarctic sea ice until now [19,27,28]. A recent study isolated six cultured phage–host systems from soil samples collected from Antarctic islands, with five phages that could infect *Pseudomonas* strains isolated from the same soil samples [29]. However, all of these phages were present in the soil in a free state, and knowledge of the phylogenetic relationships, the genetic characteristics of these phages and the interactions between phage and host is still lacking [29]. In recent years, global temperatures have risen, particularly in polar regions (polar amplification), and polar virus communities may also be affected by environmental changes, so it is important to increase the level of research on viruses in the polar areas to understand the novelty, diversity, and potential links between polar viruses and phages in other regions. The variety of the genus *Pseudomonas* and its phages makes it an excellent model for studying the interaction relationship between polar phages and their hosts at the level of individual bacteria or populations in different ecosystems [5]. An increasing number of bacteriophages infected with the *Pseudomonas* species have been isolated, and at the time of writing (February 2023), the NCBI virus had 242 determined genome sequences of phages that infect *Pseudomonas* species, revealing the tremendous diversity of *Pseudomonas* phages. These phages from NCBI RefSeq are widely distributed around the globe, but as we know, none of the *Pseudomonas* phages from the Antarctic or Arctic have been included in the NCBI RefSeq database until now.

Whether or not there are significant differences in the composition of virus communities in polar regions and lower latitudes and differences in the composition of virus communities between the Antarctic and Arctic has attracted the attention of many researchers. Despite their geographical distance, many studies have shown that the compositions of virus communities at the North and South Poles correlate. In 2015, a metagenomic analysis of polar freshwater DNA viruses showed that the taxonomic composition of the Arctic viral community was significantly different from that of viral groups in freshwater samples from lower latitudes but similar to that of groups in Antarctic freshwater and showed some viral lineages with a bipolar distribution [10]. In 2018, a study on the deep sea of Antarctica showed that the Prydz Bay virus group is similar to the Arctic freshwater virus Group and the Arctic Ocean virus group [30]. A study of metagenomic data from Antarctic animal feces and Arctic frozen soil suggested that these viral communities are genetically and functionally correlated [20]. However, despite the several metagenomic studies suggesting that the bipolar viral communities are linked, there is no direct evidence for this idea—no culturable viral–host system distributed in the North and South Poles has been isolated and characterized.

In this study, we report the isolation and characterization of a novel *Pseudomonas* lysogenic bacteriophage, vB_PaeM-G11, crossing the Antarctic and Arctic. We describe the morphological features, kinetics of phage replication, genomics, structural proteomes, and phylogenetic characteristics of the vB_PaeM-G11 phage. We demonstrate that vB_PaeM-G11 represents a novel lineage of *Autographiviridae* with five uncultivated phages from metagenomics. We also discuss the global distribution of the phage vB_PaeM-G11 and its host bacteria.

## 2. Results and Discussion

### 2.1. Biological Characteristics of Pseudomonas Phage vB_PaeM-G11

vB_PaeM-G11 was induced from the *Pseudomonas* sp. D3 (isolated from Antarctic soil in 2020), and it formed clear phage plaques on the lawn of the susceptible *Pseudomonas* sp. G11 (isolated from Arctic soil in 2018), demonstrating its high lytic capacity (Figure 1A). Considering that vB_PaeM-G11 could be integrated into the chromosome of lysogenic hosts, this lytic capacity suggests that vB_PaeM-G11 can adopt different survival strategies in the face of different host bacteria. Transmission electron microscope (TEM) observations showed that vB_PaeM-G11 has a spherical head with a diameter of about 50 nm (Figure 1B).

The experiments of phage adsorption, one-step growth, temperature, and pH sensitivity demonstrated the biological characteristics of vB_PaeM-G11. Phage adsorption curves showed that approximately 80% of vB_PaeM-G11 phages attach to the host cells after 10 min, rising to 92% at 20 min (Figure 1C). The one-step growth curve demonstrates that the latent period of vB_PaeM-G11 is about 40 min, the outbreak period is 10 min, and the burst size of one lytic cycle is about 60 pfu per infected cell (Figure 1D). To determine the change in the amplification rate of phage vB_PaeM-G11 at colder temperatures, the infected sensitive bacteria were added in an R2A broth pre-cooled to 4 °C and incubated at 4 °C with shaking, and the other conditions were consistent with the experiment at 20 °C. As expected, the result showed that when the temperature was reduced to 4 °C, the latent period of the phage vB_PaeM-G11 was substantially prolonged as the growth of the sensitive host was significantly reduced, and no lysis of the sensitive host bacteria was observed within four hours of consecutive sampling, with subsequent experiments suggesting that initial host bacterial lysis may have occurred after more than 12 h of incubation. This result implies that the lytic activity of phage vB_PaeM-G11 will increase with temperatures rising at both poles of the globe. The temperature stability experiment showed that phage vB_PaeM-G11 was stable in the range of 4–40 °C (*p*-values = 0.84); phage activity decreased sharply when the temperature exceeded 50 °C and was utterly inactive when the temperature reached 60 °C (Figure 1E). The pH stability experiment showed that vB_PaeM-G11 has a relatively broad pH tolerance and was stable at pH 4–11 (*p*-values = 0.075) (Figure 1F).

### 2.2. Genomic and Structural Proteome Characteristics of vB_PaeM-G11

The genome of the phage vB_PaeM-G11 consists of 48,000 bp linear double-stranded DNA molecules with a G + C content of 54.01%, which is lower than that of the host bacteria *pseudomonas* sp. D3 (60.63%), and no tRNA genes.

To further confirm the integration of vB_PaeM-G11 within *Pseudomonas* sp. D3, a comparison using BLASTn revealed an exact match between the genome of vB_PaeM-G11 and that in the corresponding region (2,846,729–2,887,529 bp) of the chromosome of *Pseudomonas* sp. D3. Phage Search Tool Enhanced Release (PHASTER) was used to identify phage sequences within the genome of *Pseudomonas* sp. D3; a total of 11 prophage regions were identified (Appendix A), where the predicted region, region 6 (Appendix A), and vB_PaeM-G11 genomes matched precisely. The percentage similarity between the vB_PaeM-G11 genomes and region 6 is 100%. We also found that the prophage vB_PaeM-G11 is flanked by two 14 bp repeat sequences (CGTATTCTGTCAGA), while the phage genomic DNA packed into the capsid has only one copy of this 14 bp sequence. Therefore, we believe that this 14 bp repeat sequence constitutes the phage attachment site (attP) in phages and the bacteria attachment site (attB) in bacteria, respectively [31].

A total of 50 open reading frames (ORFs) were predicted to be in the genome of vB_PaeM-G11, all genes were found to be arranged in the same direction on the sense strand, and 33 ORFs were found to have similarity with other known genes. ORF 1 encodes an integrase. ORFs 2, 6, and 39 are auxotrophic genes. ORFs 8, 22, and 28 encode proteins involved in transcription and translation. ORFs 9, 11, 16, 19, 20, 23, 26, and 30 encode DNA replication and metabolism proteins. ORFs 17 and 27 encode regulatory factors. ORFs 21, 44, and 46 encode proteins responsible for host cleavage. ORFs 32, 33, 34, 35, 36, 37, 38, 40, 41, 42, 43, 45, and 48 encode phage packaging and structural proteins. ORFs 3, 4, 5, 7, 10, 12, 13, 14, 15, 18, 24, 25, 29, 31, 47, 49, and 50 encode hypothetical proteins (Figure 2 and Table 1).

ORF 1 encodes an integrase of vB_PaeM-G11 that catalyzes the specific integration and excision of prophages from their bacteriophage host chromosomes, and the presence of this enzyme confirms the lysogenic mode of the presence of vB_PaeM-G11 in their native habitat [32].

ORF 2 is annotated to encode the bacterial outer membrane lipoprotein RcsF, a component of the Rcs signaling system. It activates the Rcs stress response system by transmitting signals from the cell surface to the histidine kinase RcsC, a bacterial Rcs system that is a stress-induced defense mechanism that controls the expression of many genes, including capsular polysaccharides, antibiotic resistance, and virulence factors involved in motility and host recognition [33]. ORF 6 encodes a 2-oxoglutarate-dioxygenase that catalyzes various reactions involving the oxidation of organic substrates using dioxygen molecules and may catalyze the hydroxylation of antibiotic peptides in prokaryotes. A member of this family, the DNA-repair protein AlkB, plays a vital role in repairing DNA damage caused by alkylating agents in E. coli [34]. ORF 39 encodes GCN5-associated n-acetyltransferases (GNATs), members of the family that are present in all living organisms and that are involved in gene regulation, transcription, protein modification, and drug resistance [35,36].

ORF 8 encodes an RNA polymerase; it can efficiently and precisely transcribe phage genes in the middle and late stages of the phage replication cycle [37]. ORF 22 encodes tRNA nucleotide transferase, which uses CTP and ATP as substrates to catalyze the transcription of CCA to the 3′ end of immature tRNA and is responsible for the maturation or repair of the 3′ end of tRNA function [38,39]. The protein encoded by ORF 28 is similar to the fusion protein 5.5/5.7 of the T7 phage and may act as a transcription effector [40].

The protein encoded by ORF 9 is similar to the 42-amino acid protein encoded by gene 1.1 in phage T7, which is rich in basic amino acids, indicating its interaction with nucleic acids, and ORF 11 encodes the ATP-dependent DNA ligase involved in DNA replication [41]. ORF 16 encodes deoxynucleoside monophosphate kinase, an enzyme required to synthesize large amounts of phage DNA rapidly [42]. ORF 19 is a T7-like gp2.5 DNA single-strand binding protein that plays a critical role in DNA replication and recombination [43]. ORF 20 encodes a phage nucleic acid endonuclease whose involvement in homologous recombination is essential in repairing DNA double-strand breaks and rescuing stalled replication forks, and it is also necessary for genetic recombination and the breakdown of host DNA [44]. ORF 23 encodes a deoxyribonucleic acid primase/helicase, which is required for the ATP-dependent unwinding of dsDNA, and it is also an essential step in DNA replication, expression, recombination, and repair. ORF 26 encodes a DNA polymerase that contains the 1X9M_A (PDB) domain and is similar to T7 phage DNA polymerase. ORF 30 encodes 5′-3′ nucleic acid exonuclease that acts as a DNA-binding protein involved in replication, recombination, and repair.

ORF 17 encodes a bacterial RNA polymerase inhibitor that includes the 4LLG_N domain of the T7 phage GP2 repressor protein, an essential inhibitor of *E. coli* RNA polymerase (RNAP). It is crucial for blocking transcription from an early promoter, contributing to the switch from host to viral RNA polymerase transcription during phage development, and is essential for the phage DNA packaging process late in infection [45,46]. The protein encoded by ORF 27 contains the PF11247 domain, a family of phage proteins that includes the phage T7 silencing protein repressor, which inhibits the gene silencing of a phage by host bacteria through the disruption of the higher-order H–NS–DNA complexes [47].

ORF 32 encodes a T7-like gp67 protein; it is found in tail-less phage particles and has been defined as a head protein. This protein may play an important role in morphogenesis, and phages mutated in this gene were found to reduce bursts of progeny [48]. ORF 33 encodes a viral particle assembly protein containing the PF11653 domain, a family of proteins representing gene product 7.3 from the T7 phage; this protein is localized in the tail and is thought to be important in viral particle assembly [48]. ORF 34 encodes portal proteins that are involved in the assembly of the phage head coat and tail and that are also crucial components of the DNA packaging complex [49]. ORF 35 and ORF 36 encode capsid assembly proteins and major capsid proteins, respectively, which synthesize and assemble the head protein shell of viruses that encapsulate their genetic material; ORF 37 and ORF 38 encode tail tube proteins that allow the phage to inject its genome into the bacterial cytoplasm without disrupting cellular integrity [49]. ORF 40, 41 and 42 encode internal viral particles corresponding to the T7 phage gp14, gp15, and gp16 proteins, respectively, which together form a hollow cylindrical core structure at the top of the portal within the capsid that acts as a transmembrane channel to deliver the genome into the cell. The protein encoded by ORF 42 also possesses lytic transglycosylase (LTase) activity, which may play a role in cell lysis, and it has been suggested that it may promote the growth and expansion of the phage at low temperatures during the early stages of infection [50,51]. ORF 43 encodes the tail fiber protein, which is located at the tail of the viral particle and is used by the phage to recognize and attach to the host bacteria [52]. ORF 45 and ORF 48 encode the small and large subunits of phage terminal enzymes, respectively, and the terminator enzymes are essential for the packaging of phage DNA [53].

ORF 44 encodes phage holin, which accumulates in plaques within the bacterial membrane and eventually forms a pore; through this pore, the endolysin is encoded by ORF 21 across the membrane to the peptidoglycan layer, thereby degrading the host cell wall and effectively completing lysis and releasing phage progeny [54]. ORF 48 encodes a phage Rz cleavage protein that contains the PF03245 domain, and the Rz protein is thought to have endopeptidase activity, which may be involved in the cleavage of oligopeptides between peptidoglycan and outer membrane Lpp lipoproteins involved in host cleavage [55].

To further verify the accuracy of the predicted results, the purified vB_PaeM-G11 particles were separated by SDS-PAGE gel. Based on the protein’s size, we identified nine SDS gel bands by Coomassie blue staining (Figure 3). All of these proteins were confirmed by the sizes of their amino acid sequences in the gel and the annotation information. These included ORF 42 (internal viral protein gp16, 146.16 kDa), ORF 38 (tail tubular protein gp12, 88.27 kDa), ORF 41 (internal virion protein gp15, 81.25 kDa), ORF 43 (tail fiber protein, 63.25 kDa), ORF 34 (portal protein, 59.01 kDa), ORF 36 (major capsid protein, 36.22 kDa), ORF 35 (capsid assembly protein, 33.24 kDa), ORF 37 (tail tubular protein gp11, 21.54 kDa) and ORF 40 (internal virion protein gp14, 19.40 kDa). The tail tubular gp11 (ORF37) and the internal viral protein gp14 (ORF40) are essentially the same sizes, and we believe they may form a band together on the gel. Except for these nine proteins, two other structural proteins, ORF32 (T7-like gp67, 9.23 kDa) and ORF33 (VirionAssem_T7, 8.94 kDa), were not significantly detected, and it is speculated that this was because of their low relative content within the phage particles. In addition, an SDS gel band between 43 kDa and 55 kDa in size did not match any of the annotated structural proteins of vB_PaeM-G11, and hence was presumably a structural protein degradation product or another unannotated structural protein.

### 2.3. Phylogenetic and Comparative Genomic Analyses

In order to determine the specific taxonomic position of vB_PaeM-G11, a genome-wide phylogenetic tree was generated using ViPTree (Figure 4A). Then, according to the genome-wide phylogenetic tree, 30 phages most closely related to vB_PaeM-G11 were selected, and a regeneration tree was constructed using ViPtree. The result indicated that the *Pseudomonas* phage vB_PaeM-G11 belongs to the *Autographiviridae* family (Figure 4B). To further determine the novelty of vB_PaeM-G11, pairwise inter-genomic distances/similarities between all identified phage genomes of *Autographiviridae* (genomes of all *Autographiviridae* phages from NCBI RefSeq) were calculated using Viral Intergenomic Distance Calculator (VIRIDIC), and a heat map was constructed (Figure 4C). The results showed that the highest genomic similarity between all identified phages belonging to *Autographiviridae* and vB_PaeM-G11 was 39.2%, which is significantly below the threshold of 95% similarity for identifying a new virus species set by the International Committee on Taxonomy of Viruses (ICTV) and also below the cut-off value of 70% similarity for identifying a new virus genus [56]. To further determine the linkage between vB_PaeM-G11 and the identified phages’ genomics, comparisons between vB_PaeM-G11 and the highest-similarity *Pseudomonas* viruses Pf1 ERZ-2017 (NC_047874.1), *Pseudomonas* phage 17A (NC_048201.1) and *Pseudomonas* phage gh-1 (NC_004665.1) were performed using DiGAlign by tBLASTx (Figure 4D). The results showed a low similarity between vB_PaeM-G11 and the *Autographiviridae* phage from NCBI RefSeq, and the homologous genes were arranged in the same order. Compared to the most closely related phage, *Pseudomonas* virus Pf1 ERZ-2017, only eight proteins showed relatively high concordance. We calculated the identity between proteins using BLASTp, and it showed a 66.52% identity for RNA polymerase, 64.85% identity for DNA ligase, 70.34% identity for DNA polymerase, 80.52% identity for portal protein, 80.12% identity for major capsid protein, 73.33% identity for tail tubular protein gp11, 68.85% identity for tail tubular protein gp12, and 73.22% identity for the terminase large subunit. Therefore, we believe that vB_PaeM-G11 represents a separate virus cluster in the *Autographiviridae* family at both the genomic and genetic levels.

To identify homologous viral genome sequences of the vB_PaeM-G11 present in the environment, we searched for vB_PaeM-G11-related DNA sequences in the NCBI *Caudoviricetes* virus and IMG/VR v4 using BLASTn (E-value < 1.00 × 10^−10^, query coverage > 25%). The result showed that only five related uncultivated viral genomes (UViGs) from the IMG/VR database were matched. Then, a total of 55 virus clusters (VC) were generated by clustering 385 genomes (379 *Autographiviridae* phages of NCBI RefSeq, five from IMG/VR, and vB_PaeM-G11) using vConTACT 2.0. The clustering results fit well with ICTV’s taxonomic results for 133 genera in the *Autographiviridae* family (Figure 5). For ease of display, the network diagram does not separately show unclustered phages and the genera under the *Autographiviridae* family that currently have only one species; instead, they are grouped under a singleton. vB_PaeM-G11 could be clustered with the genomes of five UViGs from IMG/VR into the single viral cluster VC-0-0, with no correlation with other phages, and the phages from cluster VC-0-0 were also aligned using DiGAlign, all showing high similarity to vB_PaeM-G11 (Appendix A).

The VICTOR service showed a high agreement with the ICTV classification at the genus level. A genome-wide phylogenetic tree was constructed by combining the *Pseudomonas* phage vB_PaeM-G11 genome with that of the high-similarity *Autographiviridae* phages calculated by VIRIDIC and that of the related UViGs from the IMG/VR database for a total of 100 bacteriophage genomes (Figure 6). The results indicated that vB_PaeM-G11 and five related UViGs from IMG/VR form a separate branch that represents a new viral cluster. In conclusion, all the results indicated that phage vB_PaeM-G11 represents a new genus of *Autographiviridae*, which we named *Fildesvirus*.

### 2.4. The Global Distribution Range of Phage vB_PaeM-G11

To further understand the worldwide distribution of vB_PaeM-G11 and members of the genus *Fildesvirus*, we searched for the sample sources of five related UViGs from the IMG/VR database. Two of these UViGs, from permafrost metagenomic data (Arctic tundra, 2013), homologous to vB_PaeM-G11 with over 95% similarity, were considered the same species of vB_PaeM-G11 according to the classification rules of ICTV [56]. The other three phage genomes were from the plant inter-root soil macrogenome (MI, USA, 2019), and they are approximately 90% similar to those of vB_PaeM-G11 (Appendix A). Meanwhile, the sensitive host of the phage vB_PaeM-G11, *Pseudomonas* sp. G11, was isolated in 2020 from the soil of Arctic Svalbard. More interestingly, the lysogenic host of the phage vB_PaeM-G11, *Pseudomonas* sp. D3, was isolated from the soil of the Fildes Peninsula (King George Island, Antarctica) in 2018 (Figure 7). The genome sequences of *Pseudomonas* sp. D3 and *Pseudomonas* sp. G11 were uploaded to Type Strain Genome Server (TYGS) (https://tygs.dsmz.de, accessed on 6 December 2022) for a whole genome-based taxonomic analysis. The results showed that both *Pseudomonas* sp. D3 and *Pseudomonas* sp. G11 were detected as potentially new species which do not belong to any species in the TYGS database—the dDDH values between them and their most relevant bacteria were all less than 70% [57]. The dDDH values of *Pseudomonas* sp. D3 and *Pseudomonas* sp. G11 and a set of type strains are shown in Appendix A (Appendix A). The phylogeny of GBDP based on the 16S rDNA gene sequence showed that *Pseudomonas Fildesensis* KG01 (soil, Antarctica King George Island) is the most closely related species to *Pseudomonas* sp. D3, and *Pseudomonas veronii* DSM 11331 (natural mineral water, France) was the most closely related species to *Pseudomonas* sp. G11 [58,59] (Appendix A). To further understand the transmission pathway of vB_PaeM-G11 between the Antarctic and Arctic, we tried to use BLASTn (E-value < 1.00 × 10^−5^, query coverage > 25%) to find homologous phages of vB_PaeM-G11 in the GOV2.0 database. The result showed that no similar viral DNA sequences to vB_PaeM-G11 were found, which may indicate that *pseudomonas* phage vB_PaeM-G11 is not transmitted between the Antarctic and Arctic through ocean currents but through other pathways.

Due to their ability to fly, migratory birds are increasingly being considered to play a potentially important role in the transmission of viruses and bacteria in polar regions [60,61]. For example, a recent study has shown that viruses carried by Antarctic seabirds have connectivity with viral communities in South America [60], and another study showed potential links between Antarctic migratory bird fecal virus communities and other environmental and biological entities around the world by comparing the viral genomes of Antarctic migratory bird feces with those of other known viruses [20]. It is reasonable to speculate that birds may cause the spread of the vB_PaeM-G11 phage between the Antarctic and Arctic. Arctic terns are strongly migratory, migrating along a convoluted route from Arctic breeding grounds to the Antarctic coast and back again about six months later [62]. A recent study of antibiotic-resistant bacteria in Arctic terns from the Svalbard Archipelago suggests the potential role of Arctic terns in the spread of multidrug-resistant bacteria in polar environments [61]. Since vB_PaeM-G11 can be present in the host genome as a prophage, following the host bacterium in lysogenic form becomes a possible option for bipolar transmission. However, the microbiomes of polar migratory birds such as Arctic terns and Antarctic polar skuas are still poorly studied, and related data in the database are still scarce, so further confirmation of the transmission of vB_PaeM-G11 between the poles still needs to be supported by more taxonomic and metagenomic data. Although it is not clear how the vB_PaeM-G11 phage achieves simultaneous distribution in the Antarctic and Arctic, the isolation and characterization of vB_PaeM-G11 distribution as the first form of phage–host system distribution in the Antarctic and Arctic provide direct evidence for the findings of previous related studies—some viruses are distributed simultaneously at both poles of the Earth, although extreme distances and other physical barriers separate these environments [10]. Furthermore, in addition to demonstrating that vB_PaeM-G11 is a bipolarly distributed phage, we isolated bacterial hosts of vB_PaeM-G11 from the Antarctic and Arctic. These findings predicted that this phage could function in both Antarctic and Arctic ecosystems in a lysic or lysogenic manner.

## 3. Materials and Methods

### 3.1. Host Bacteria Culture and Phage Induction

The 95 strains of the genus *Pseudomonas* selected for the experiment were isolated from the Antarctic and Arctic in the last ten years (Appendix A), and all strains were obtained in pure culture by picking single colonies from dilution and spread plates and conserved at the China Center for Type Culture Collection (CCTCC). These bacteria were activated and inoculated in 7 mL of R2A Broth (BD Difco™, Loveton Circle Sparks, MD, USA) and incubated at 20 °C until OD600 = 0.8, then centrifuged at 12,000× *g* for 2 min at 4 °C and resuspended in 1 mL of fresh R2A Broth. After resuspension, the bacteria were laid flat on the bottom of the Petri dish with UV irradiation (120 μw/cm^2^) for 2 min, avoiding light. Then, 5 mL of R2A Broth was added to the dish and placed in a dark incubator at 20 °C for 48 h. The bacteria were collected and centrifuged at 12,000× *g* for 2 min at 4 °C; the supernatant passed through a 0.22 μm pore-sized membrane (Millipore, Billerica, MA, USA), and it was stored at 4 °C for use.

### 3.2. Phage Isolation, Amplification, and Purification

Phages were isolated by the double-layer agar method. After 1 mL of each supernatant induced by UV was mixed, 100 μL of each supernatant was separately mixed with 500 μL of the mid-log bacterial culture of the 95 *Pseudomonas* strains and incubated for 2 h at 20 °C. After incubation, the mixture was mixed with 5 mL of the R2A medium containing 0.5% agar (cooled to 50 °C) and poured onto the bottom agar layer. This was allowed to solidify, incubated upside down in an incubator for one week at 20 °C, and observed every day during this time. To identify the lysogenic host of the phage, when phage plaques were observed on the lawn of the *Pseudomonas* strains, each supernatant induced by UV was incubated separately with the culture of sensitive host bacteria and then observed for phage plaque using the double-layer plate method. The phages were purified by picking single-phage spots seven times in succession and then storing the purified phages in the SM buffer (10 mM Tris-HCl (pH = 7.5), 100 mM NaCl, and 10 mM MgSO_4_) at 4 °C [63].

A purified phage from a single-phage spot was incubated at a multiplicity of infection (MOI) of 0.001 in 5 mL of the sensitive host bacterial culture (OD600 = 0.8) for 2 h at 20 °C, then transferred to 400 mL of R2A Broth and incubated for 24 h at 20 °C with oscillation (160 rpm). The infected bacterial culture was centrifuged at 7000× *g* for 20 min to precipitate bacterial debris, and then the phage lysate supernatant was precipitated with polyethylene glycol 8000 (10%) and NaCl (0.6%) for more than 24 h at 4 °C. At the end of the incubation, the white precipitate containing a crystalline virus was centrifuged at 8000× *g* for 20 min at 4 °C, and the precipitate was resuspended in the SM buffer. After the addition of 1 M KCl (Aladin, Shanghai, China), the mixture was incubated on ice for 20 min and centrifuged at 12,000× *g* for 10 min at 4 °C to precipitate the PEG8000 and aspirate the supernatant containing the phage [64]. Cesium chloride (Aladin, Shanghai, China) was added at a rate of 0.7 g per mL of phage supernatant and purified by ultracentrifugation at CsCl density gradients of 35,000 rpm (Optima XE-100, SW41 Ti rotor, Beckman, Brea, CA, USA) for 24 h at 4 °C [63]. Then, visible viral bands were collected and dialyzed using a 100 KDa cut-off in a centrifugal ultrafiltration tube (Millipore, Billerica, MA, USA) with the SM buffer. The purified phage was stored at 4 °C for use.

### 3.3. Transmission Electron Microscopy

Purified phage pellets were negatively stained with 1% (*w*/*v*) uranyl acetate, then stained particles were observed at 100 kV using the JEOL JEM-1400plus transmission electron microscope (TEM) [8].

### 3.4. Phage Biology Experiments

The data analysis and presentation of results were achieved by GraphPad Prism 8, and a one-way analysis of variance (ANOVA) was used to identify the significant differences. All the results shown were average values from the triplicate experiments, and the error bars indicate standard deviation (SD).

#### 3.4.1. Phage Adsorption

The phages and host bacteria culture (OD600 = 0.8) were mixed at a multiplicity of infection (MOI) of 0.01 and incubated at 20 °C, and then 200 μL of the mixture was taken at specific time points: 0, 5, 10, 15, 20, 25, and 30 min. After centrifuging at 12,000× *g* for 2 min, 100 μL of the supernatant was diluted in 0.9 mL of R2A Broth (cooled to 20 °C), and the number of unattached phages in the supernatant was counted by the double-layer method [65].

#### 3.4.2. One-Step Growth

Phages and host bacteria were mixed at an MOI of 0.001 and incubated for 20 min at 20 °C to allow adsorption. The suspensions were centrifuged at 10,000× *g* for 2 min at 4 °C, and the bacterial precipitates were washed in R2A Broth three times. Finally, they were suspended in 50 mL of R2A Broth pre-cooled to 20 °C or 4 °C and incubated at 20 °C or 4 °C with oscillation (160 rpm). Samples were taken every 10 min, and the phage titers were determined by the double-agar method [65].

#### 3.4.3. Temperature and pH Tolerance Range of Phage

In terms of temperature, the phages present in the SM buffer were incubated at 4, 10, 20, 30, 40, 50, 60, 70, 80, 90, and 100 °C for 1 h, and then the titer of phage was determined by the double-layer agar method. In terms of pH, the phages were inoculated in the SM buffer with the pH being adjusted to 2, 3, 4, 5, 6, 7, 8, 9, 10, 11, and 12 with 1 M HCl or 1 M NaOH and incubated for 1 h at 20 °C. The titer of the phages was determined by the double-layer agar method [66].

### 3.5. Phage and Bacterial DNA Extraction and Sequencing

Purified genomic phage DNA was extracted by MiniBEST Viral RNA/DNA Extraction Kit Ver.5.0 (Takara, Kyoto, Japan). The extracted phage genomic DNA was sequenced by AUGCT (Beijing, China). The phage genomic DNA was first randomly broken into small fragments of approximately 350 bp, then DNA fragments of the desired length were collected, and specific connectors were selected using the NEB standard library building kit. Library preparation was performed on this sample, library fragment size was detected using aglent 2100, and qPCR detected the library concentration. After passing the library check, PE 2 × 150 sequencing was performed using Illumina NovaSeq, and library-built splice sequences were removed using a cut adapter software. Quality control was performed using FastQC to obtain available reads, Spades was used for sequence splicing, and assembled sequences were evaluated using the PhageTerm software to perform software correction and find features for assembled sequences [67,68].

The genomic DNA of *Pseudomonas* strains was extracted by using a genomic DNA extraction kit (Qiagen, Hilden, Germany) following the manufacturer’s instructions. The DNA was checked for purity on Nanodrop (Thermo Scientific, Waltham, MA, USA), and DNA concentrations were measured using Qubit 3.0 Fluorometer (Life Technologies, Carlsbad, CA, USA). For long-read sequencing, the libraries were prepared using the SQK-LSK109 ligation kit following the manufacturer’s instructions. The purified library was loaded onto R9.4 FLO-PRO002 flow cells and sequenced using a PromethION sequencer (Oxford Nanopore Technologies, Oxford, UK) with 48 h runs at Wuhan Benagen Technology Company Limited (Wuhan, China). A base calling analysis of raw data was performed using the Oxford Nanopore GUPPY software (v5.0.16). Reads with a quality score lower than 7 were discarded. The reads were aligned to the bacterial strain reference genome, using minimap2 (v.2.24) and Samtools (v.1.9) to filter out the bacterial strain reads and extract the unmatched BAM file [69,70], and then bam2fastx was used to convert the fq-sequence. The final reads were annotated using Karken2 (v.2.1.2) and Bracken (v.2.6.2) and bridged to the contig using Flye (v2.9) [71,72,73].

### 3.6. Genomic and Structural Proteome Characteristics

The putative open reading frames (ORFs) of the phage were predicted by Prokka [74], and all ORFs were annotated by HHpred with default parameters based on four databases (PDB_mmCIF70_12_Aug, Pfam-A_v35, COG_KOG_v1.0, and NCBI_Conserved_Domains (CD)_v3.19) [75]. Genome mapping was carried out using Proksee (https://proksee.ca/, accessed on 6 December 2022), and the tRNAscan-SE program predicted tRNA sequences [76]. The prediction of *Pseudomonas* lysogenic phages was completed by the online server of PHASTER [77]. The structural protein of phage vB_PaeM-G11 was identified by 12% SDS-PAGE. An amount of 20 μL of purified phage particles were suspended in the SDS-PAGE protein loading buffer (Beyotime, Shanghai, China) and heated for 5 min at 95 °C, and then, the structural proteins were separated by SDS-PAGE gels and stained by Coomassie blue staining [78]. The protein marker chosen for the experiment was the PageRuler pre-stained protein ladder (Thermo Fisher Scientific, Waltham, MA, USA).

### 3.7. Phylogenetic and Comparative Genomic Analysis

The generation of a proteomic tree of viral genome sequences based on genome-wide similarity calculated by tBLASTx using the ViPTree server [79], viral genome sequences and taxonomic information of viruses and their hosts were based on the virus–host DB [80]. Thirty phages related to vB_PaeM-G11 were selected, and a regenerative proteomic tree was constructed using ViPTree to determine the taxonomic range of vB_PaeM-G11 at the family level. The genomic nucleotide similarity between vB_PaeM-G11 and all identified phage genomes belonging to *Autographiviridae* was calculating using Viral Intergenomic Distance Calculator (VIRIDIC) to determine their taxonomic relationships [81]. Genomic comparisons between vB_PaeM-G11 and the most closely related phage were performed using DiGAlign (https://www.genome.jp/digalign/, accessed on 2 December 2022).

vConTACT2 is a tool used to perform guilt-by-contig-association classification of phage genome sequence data [82]. We used BLASTn (E-value < 1.00 × 10^−10^, query coverage > 25%) to search for 82,577 phages of the class *Caudoviricetes* downloaded from the NCBI Nucleotide database (https://www.ncbi.nlm.nih.gov/nucleotide, accessed on 1 November 2022) and used the same criteria for database IMG/VR v4 [83]. The phages related to vB_PaeM-G11 were analyzed together with 379 *Autographiviridae* phages from NCBI-RefSeq and vB_PaeM-G11. A comparison of the proteins of all phages using Diamond BLASTp was carried out (E-value ≤ 1.00 × 10^−10^, coverage ≥ 85%, and amino acid identity ≥ 80%) [84]. Protein clustering using the Markov clustering algorithm (MCL) was performed to generate protein clusters (PC), and ClusterONE was used to generate virus clusters (VC) based on the protein clustering relationships [85,86]. Finally, the clustering scores were calculated by Vcontact2, and the generated network graphs were visualized by Cytoscape 3.9.1 [87].

Using Virus Classification and Tree Building Online Resource (VICTOR), a method for the genome-based phylogeny and classification of prokaryotic viruses [88], a vB_PaeM-G11 genome-wide phylogenetic tree was constructed using the VICTOR web service (https://victor.dsmz.de, accessed on 8 November 2022). The nucleotide sequences were compared using the genome explosion distance phylogeny (GBDP) method in the settings recommended for prokaryotic viruses [89], and the branch length was magnified by the distance formula d0 according to GBDP. The resulting tree file was visualized by ggtree [90].

### 3.8. The Global Distribution of vB_PaeM-G11 Phage

The source of vB_PaeM-G11-related phage genomes from IMG/VR was searched on the JGI website (https://img.jgi.doe.gov/cgi-bin/vr/main.cgi, accessed on 15 October 2022). The similarity between the genome sequence of phage vB_PaeM-G11 and the related phages’ genome sequences was calculated using OrthoANI [91]. The genome sequence data of the host bacteria were uploaded to the Type (Strain) Genome Server (TYGS) (https://tygs.dsmz.de, accessed on 6 December 2022) for a whole genome-based taxonomic analysis [92]. The analysis also made use of recently introduced methodological updates and features [57]. Digital DDH values and confidence intervals were calculated using the recommended settings of GGDC 3.0 [57,89]. Type-based species clustering using a 70% dDDH radius around each of the closest type strain genomes was carried out as previously described. The homologous phage of vB_PaeM-G11 in Global Ocean was searched in the Global Ocean Viromes 2.0 (GOV 2.0) database [18].

### 3.9. Nucleotide Sequence Accession Numbers

The complete genome of *Pseudomonas* sp. D3, *Pseudomonas* sp. G11, and the vB_PaeM-G11 phage, can be accessed under the GenBank accessions CP119772, CP120377 and OQ622254, respectively.

## 4. Conclusions

*Pseudomonas* plays a pivotal role in polar ecosystems. In this study, we identified and characterized a polar-derived *Pseudomonas* virus, a Lysogenic *Pseudomonas* bacteriophage crossing the Antarctic and Arctic, providing definitive evidence that some phages have bipolar distribution. We described the biological characteristics, genome, phylogeny, and global distribution of this phage and demonstrated that vB_PaeM-G11 represents a new lineage of viruses in the natural environment and could represent a novel viral genus of *Autographiviridae*, named *Fildesvirus* here. In general, we described the isolation of a cultured phage–host system from the soil of polar regions, and the study of this new cultured phage–host system provides valuable information for understanding the biological characteristics and ecological functions of polar phages and the interaction relationship between polar phages and their hosts and helps to predict and analyze viral sequences from metagenomic databases. In the future, studies of the functional genes of phages and their hosts will help researchers to understand their role in the evolution and adaptive capacity of polar microorganisms; further study of the metagenetics associated with polar migratory birds may help to elucidate the linkages between bipolar viruses and the possible pathways of bipolar virus transmission.

## Figures and Tables

**Figure 1 ijms-24-07662-f001:**
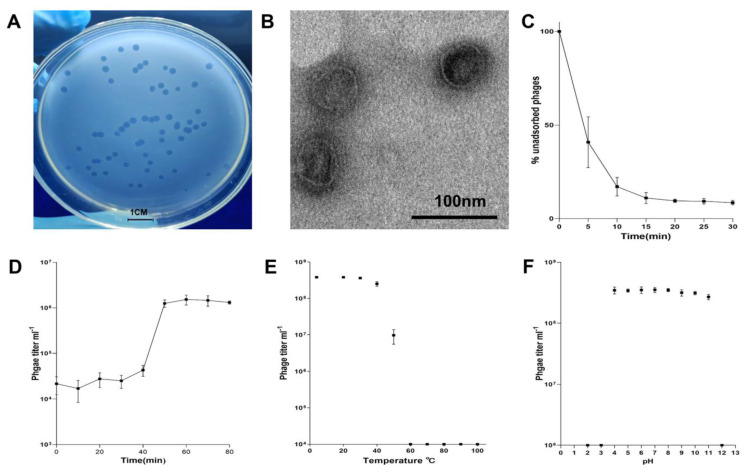
(**A**) Phage plaques formed by vB_PaeM-G11 on the lawn of its sensitive strain, *Pseudomonas* sp. G11. (**B**) Transmission electron micrograph (TEM) of phage vB_PaeM-G11. (**C**) Adsorption curves of phage vB_PaeM-G11 on *Pseudomonas* sp. G11. (**D**) One-step growth curve of vB_PaeM-G11. (**E**) Temperature adaptation range of vB_PaeM-G11. (**F**) pH adaptation range of vB_PaeM-G11.

**Figure 2 ijms-24-07662-f002:**
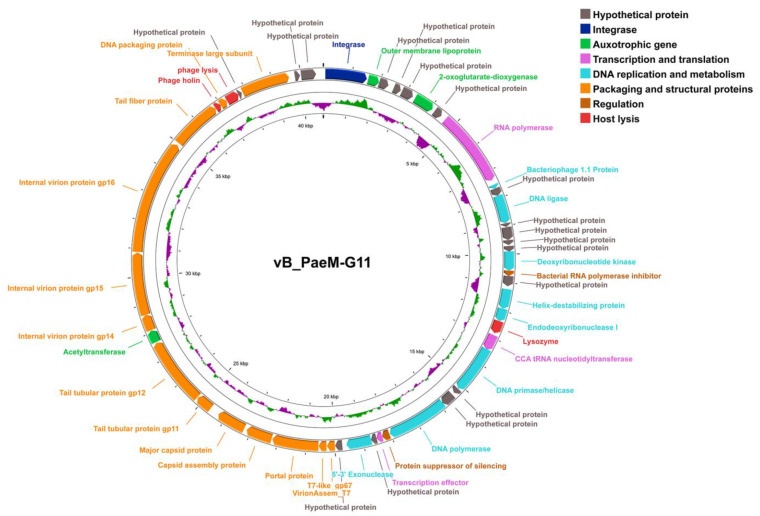
Circular diagram of functional assignments of the *Pseudomonas* phage vB_PaeM-G11 genome. The outermost circle represents the 50 ORFs contained by phage vB-PaeM-G11. Arrows represent the direction of gene transcription, and different colors indicate genes with different functions: integrase (dark blue); auxiliary metabolic genes (green); transcription- and translation-related genes (purple); DNA replication and metabolism genes (light blue); transcriptional regulatory genes (brown); packaging and structural protein genes (orange); host cleavage genes (red); and hypothetical protein (gray). The GC skew of the genome sequence is indicated by the internal purple or green histograms.

**Figure 3 ijms-24-07662-f003:**
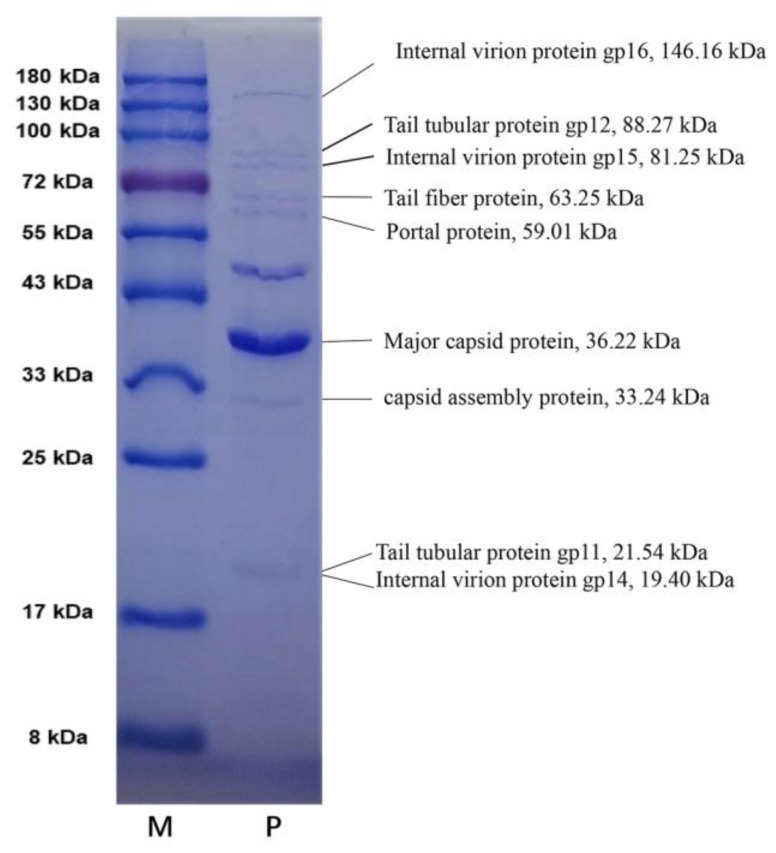
Analysis of 12% SDS-PAGE for phage vB_PaeM-G11 structural proteins. M represents the protein molecular weight marker. P represents the structure protein solution of vB_PaeM-G11. On the right are the descriptions of the molecular sizes based on the protein sequences and their possible functions.

**Figure 4 ijms-24-07662-f004:**
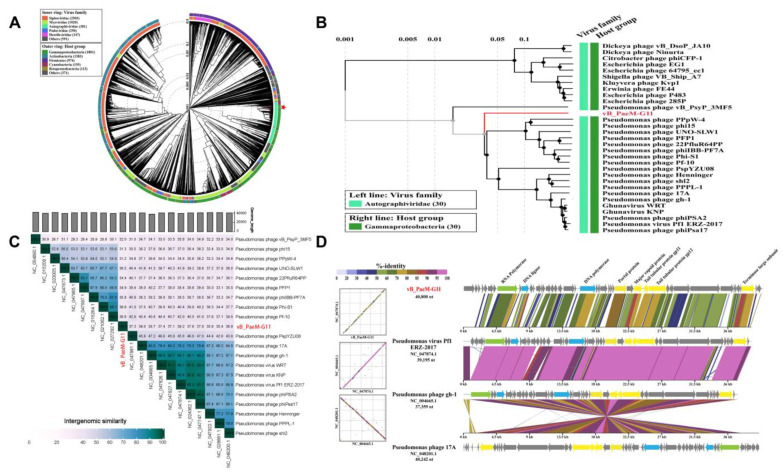
Phylogenetic and comparative genomic analysis of vB_PaeM-G11. (**A**) Proteomic tree of viral genome sequences based on the whole genome represented in the circular view, and a red pentagram marks vB_PaeM-G11. (**B**) Regenerated proteomic tree of vB_PaeM-G11 and 30 closely related phage genome sequences. (**C**) Intergenomic similarity between vB_PaeM-G11 and the most similar phage calculated using VIRIDIC. The heat map indicates the similarity between genomes; for display purposes, only the top 20 high-similarity species are shown. (**D**) Genome alignment between vB_PaeM-G11 and the high-similarity phage from NCBI RefSeq. High-similarity regions detected by tBLASTx are color-coded in the alignment based on the reported identity %. Dot plots summarizing these high-similarity regions are shown beside the alignment.

**Figure 5 ijms-24-07662-f005:**
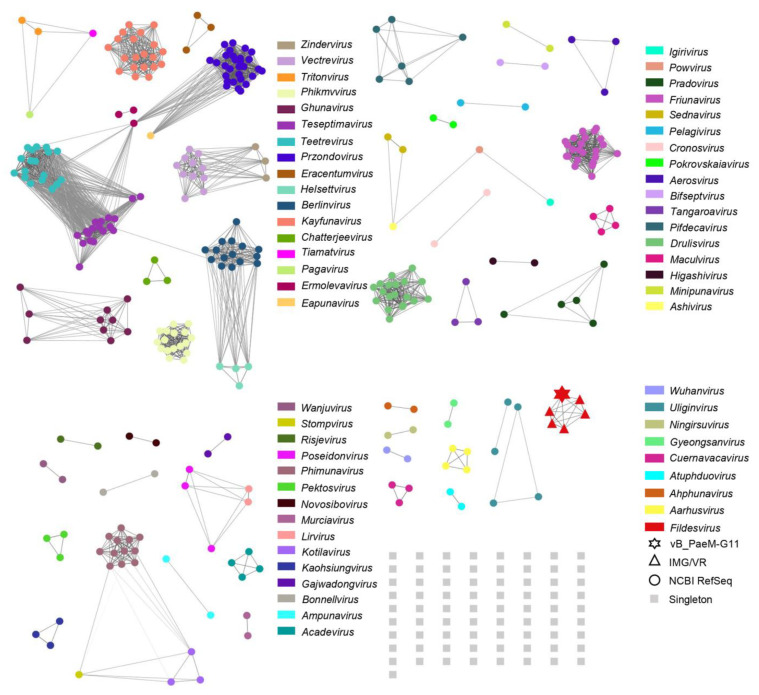
Proteome-based similarity network showing all vB_PaeM-G11 *Autographiviridae* phages from the NCBI RefSeq, and five related UViGs from the IMG/VR database. Each node represents a single phage genome, and the edge represents similarity scores between proteomes of related phages. Different colors indicate the clustered viruses. Circles represent phages from the NCBI RefSeq database, the hexagon represents vB_PaeM-G11, and triangles indicate UViGs from the IMG/VR database. Gray squares represent singleton nodes. The network was visualized using Cytoscape 3.9.1.

**Figure 6 ijms-24-07662-f006:**
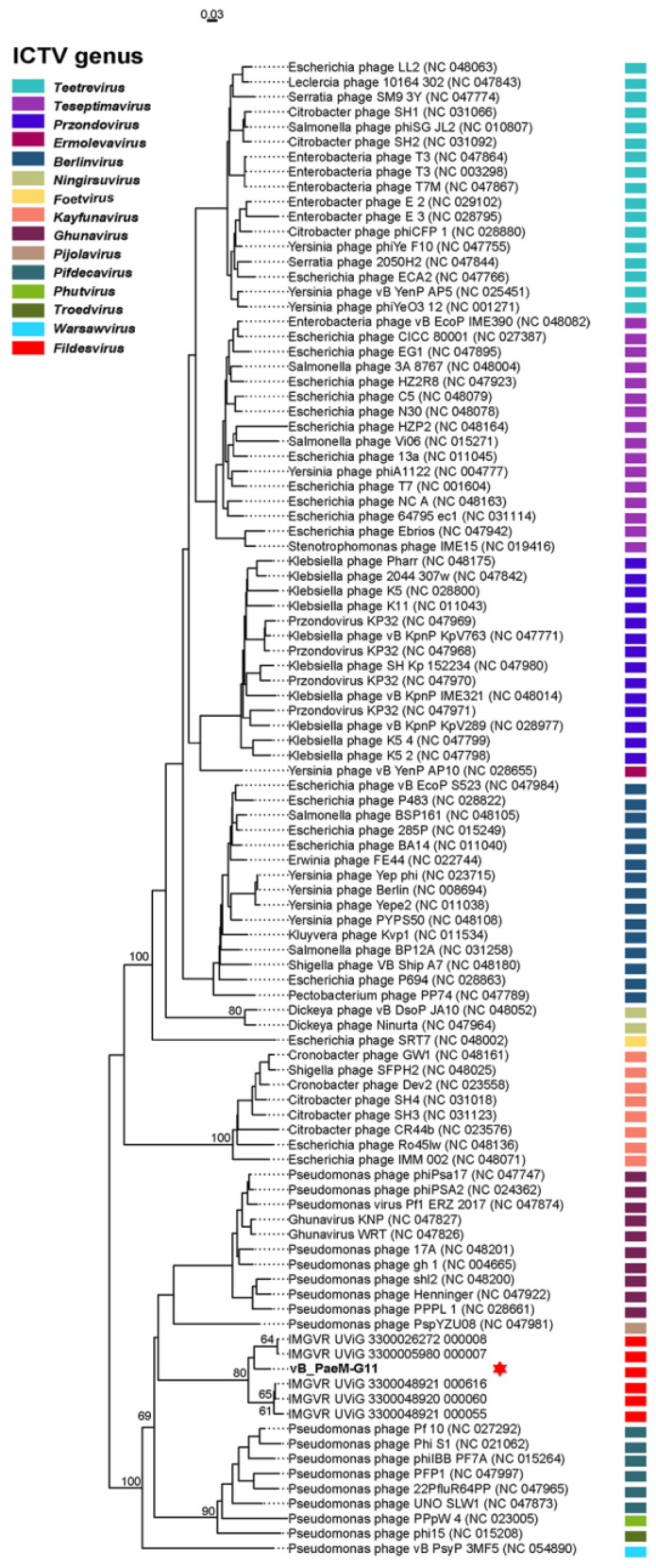
Genome-wide phylogenetic tree constructed by VICTOR with the formula d0. The phylogenic tree consists of 100 phage genomes (vB_PaeM-G11, 94 *Autographiviridae* phages from NCBI RefSeq and five related UViGs from IMG/VR). Each unique color indicates each ICTV genus. The vB_PaeM-G11 genus is shown in red, and a red hexagon marks vB_PaeM-G11. Bootstrap values of ≥50 are shown.

**Figure 7 ijms-24-07662-f007:**
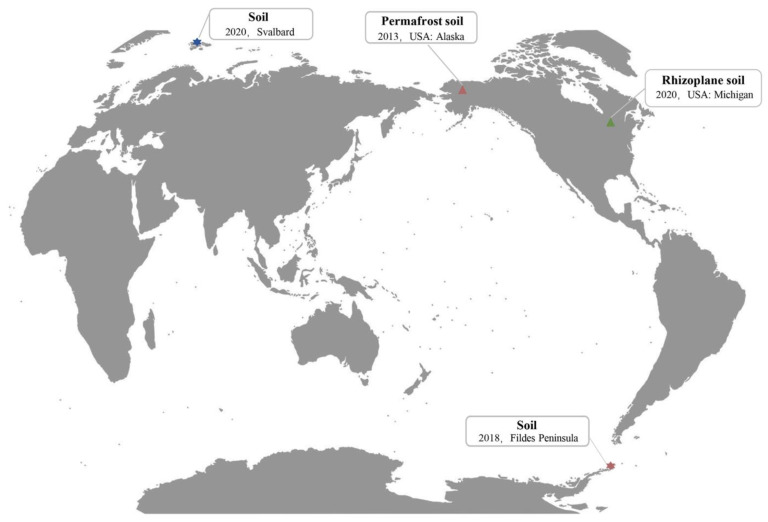
Biogeography of the *Fildesvirus*. A blue hexagon represents *Pseudomonas* sp. G11. A red hexagon represents *Pseudomonas* sp. D3. Two UViGs (no. IMGVR_UViG_3300005980_000007 and IMGVR_UViG_3300026272_000008) that have over 95% similarity to vB_PaeM-G11 are represented by red triangles. The other three UViGs (no. IMGVR_UViG_3300048920_000060, IMGVR_UViG_3300048921_000055, and IMGVR_UViG_3300048921_000616) are represented by green triangles. The labels show the ecosystem type, sample collection date, and geographic location of the microbiome.

**Table 1 ijms-24-07662-t001:** Genomic annotation of *Pseudomonas* phage vB_PaeM-G11.

ORF	Size (aa)	Prediction Function	Conserved Domains	E-Values
1	486	Integrase	COG4973	1.40 × 10^−26^
2	131	Outer membrane lipoprotein	6T1W_C (PDB)	1.20 × 10^−17^
6	238	2-oxoglutarate-dioxygenase	6FXR_A (PDB)	1.00 × 10^−17^
8	897	RNA polymerase	1MSW_D (PDB)	3.30 × 10^−157^
9	47	Bacteriophage 1.1 Protein	PF08200.14 (Pfam)	1.70 × 10^−9^
11	331	DNA ligase	1A0I_A (PDB)	9.30 × 10^−36^
16	222	Deoxyribonucleotide kinase	1DEK_A (PDB)	7.40 × 10^−17^
17	63	Bacterial RNA polymerase inhibitor	4LLG_N (PDB)	2.60 × 10^−25^
19	230	Helix-destabilizing proteins	1JE5_B (PDB)	5.30 × 10^−34^
20	149	Endodeoxyribonuclease I	1M0D_B (PDB)	2.90 × 10^−28^
21	148	Lysozyme	1LBA_A (PDB)	1.90 × 10^−13^
22	184	CCA tRNA nucleotidyltransferase	KOG2159	2.40 × 10^−22^
23	563	DNA primase/helicase	6N7I_C (PDB)	1.20 × 10^−64^
26	715	DNA polymerase	1X9M_A (PDB)	1.00 × 10^−68^
27	88	Protein suppressor of silencing	PF11247.11 (Pfam)	1.20 × 10^−29^
28	69	Transcription effector	5LGM_A (PDB)	1.50 × 10^−38^
30	292	5′-3′ exonuclease	6C33_A (PDB)	3.80 × 10^−28^
32	90	T7-like_gp67;	PF17570.5 (Pfam)	6.00 × 10^−23^
33	89	VirionAssem_T7	PF11653.11 (Pfam)	1.40 × 10^−28^
34	541	Portal protein	6R21_J (PDB)	5.50 × 10^−60^
35	306	capsid assembly protein	PF05396.14 (Pfam)	9.00 × 10^−28^
36	341	Major capsid protein	3J7W_D (PDB)	5.00 × 10^−30^
37	192	Tail tubular protein gp11	6R21_X (PDB)	2.00 × 10^−41^
38	794	Tail tubular protein gp12	6R21_b (PDB)	1.00 × 10^−105^
39	145	Acetyltransferase	3KKW_A (PDB)	9.00 × 10^−14^
40	188	Internal virion protein gp14	7EY7_A (PDB)	8.50 × 10^−31^
41	734	Internal virion protein gp15	7K5C_I (PDB)	6.30 × 10^−117^
42	1349	Internal virion protein gp16	7K5C_G (PDB)	5.60 × 10^−122^
43	589	Tail fiber protein	7EY9_r (PDB)	7.30 × 10^−47^
44	67	Phage holin	PF10746.12 (Pfam)	9.70 × 10^−21^
45	85	DNA packaging protein	PF11123.11 (Pfam)	3.50 × 10^−22^
46	150	Phage lysis	PF03245.16 (Pfam)	3.70 × 10^−16^
48	569	Terminase large subunit	8DGC_G (PDB)	5.30 × 10^−55^

## Data Availability

The original contributions presented in the study are included in the article/Appendix A, and further inquiries can be directed to the corresponding author.

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
