# Peer review of "A Pseudomonas Lysogenic Bacteriophage Crossing the Antarctic and Arctic, Representing a New Genus of Autographiviridae"

_ijms, 2023, doi:10.3390/ijms24087662_

Round 1
Reviewer 1 Report
In the study “A Pseudomonas lysogenic bacteriophage crossing the Antarctic and Arctic, representing a new genus of Autographiviridae.” Liu et.al, isolated and characterized a lysogenic phage of Pseudomonas sampled from the Arctic and Antarctic regions. The new phage was characterized as vB_PaeM-G11 named Fildesvirus, belonging to the genus Autographiviridae. Genome sequencing and comparison were performed to identify the relationship between the isolated phage and other viral genomes retrieved from various databases.
The current study addresses the phylogenetic relationship of lysogenic phage and the evolution of the phage-host defense mechanism and HGT of the host bacteria. The manuscript is written nicely and is easy to read.
Please consider reducing the text mentioned below.
#Major comments.
Line 126 – Please explain “exact match” here. What is the percentage similarity between the phage and the phage region within the genome?
Figure 2 – Would be nice if authors reconsider the colors used here.
Lines 136-228 – Consider reducing the text here and converting it into a very informative table for the readers.
Lines 339-340 – Rephrase the sentence.
Lines 351 – If not through ocean currents, please clarify the possible pathways.
Line 409 – Change to “microscopy” (to denote the technique used here)
Author Response
Response to Reviewer 1 Comments
Point 1: Line 126 – Please explain “exact match” here. What is the percentage similarity between the phage and the phage region within the genome?
Response 1: Line 166 – The percentage similarity between the phage and the phage region within the genome of Pseudomonas sp.D3 is 100%.
Point 2: Figure 2 – Would be nice if authors reconsider the colors used here.
Response 2: We have changed the colors used in Figure 2.
Point 3: Lines 136-228 – Consider reducing the text here and converting it into a very informative table for the readers.
Response 3: A table of this section was previously produced and included in the supplementary material, which we have now included in the main text, as shown in Table 1.
Point 4: Lines 339-340 – Rephrase the sentence.
Response 4: Lines 370-372 – In the revised manuscript, this sentence has been rewritten.
Point 5: Lines 351 – If not through ocean currents, please clarify the possible pathways.
Response 5: We conjecture that the spread of vB_PaeM-G11 between the Antarctic and Arctic may be carried by birds (e.g., Arctic terns), but there is insufficient evidence to support this speculation.
Point 6: Line 409 – Change to “microscopy” (to denote the technique used here)
Response 6: Line 468 – Microscope" has been replaced with "microscopy" in the revised manuscript.
Reviewer 2 Report
The manuscript, entitled “ A Pseudomonas lysogenic bacteriophage crossing the Antarctic and Arctic, representing a new genus of Autographiviridae. " This work is merited for publication in International Journal of Molecular Sciences after some major modification. So, I have some points that may help to improve the work as follows:
1-Abstract is good but need more explain about the main aim of work
2- The introduction should be extended to discuss the hypothesis and research questions in details. Additionally, the introduction should cover the recent literature related to this subject.
3- Material and methods
The methodologies should be explained in details so that the results are reproducible.
4-Results
The results are clear and important.
5-Discussion
The discussion section still needs improvement, and should be linked to the findings of the previous reports on this topic.
6- The conclusion
A section for conclusions need more explain and should include the most significant findings and future works only.
7- English writing should be checked by a native English-speaking expert.
Author Response
Response to Reviewer 2 Comments
Point 1: Abstract is good but need more explain about the main aim of work
Response 1: Lines 9-10 – In the revised manuscript, we have made some additions and changes to the abstract section.
Point 2: The introduction should be extended to discuss the hypothesis and research questions in details. Additionally, the introduction should cover the recent literature related to this subject.
Response 2: Lines 62-112 – In the revised manuscript, we have partially supplemented and expanded the introductory section and covered as much of the relevant and up-to-date literature as possible.
Point 3: The methodologies should be explained in details so that the results are reproducible.
Response 3: line 429-432, line 444, line461, line 473-476, line 481, line 535-536 – We have checked the "Materials and Methods "section and added a few details, and it may now be clear and easier to understand.
Point 4: The results are clear and important.
Response 4: Thank you, we are delighted to receive your compliments.
Point 5: The discussion section still needs improvement, and should be linked to the findings of the previous reports on this topic.
Response 5: Lines 389-415 – In the revised manuscript, we have made some additional changes and corrections to the results and discussion section and linked them to other previous research related to this work where possible.
Point 6: A section for conclusions need more explain and should include the most significant findings and future works only.
Response 6: Lines 583-598 – In the revised manuscript, we have made some additions and corrections to the conclusion section
Point 7: English writing should be checked by a native English-speaking expert.
Response 7: Thank you for your valuable and thoughtful comments. We have carefully checked and improved the English writing in the revised manuscript.
Reviewer 3 Report
Dear Editors and authors,
1-The abstract of the manuscript is very weak and needs to be supported by adding some of the obtained results.
2-The scientific names of living organisms are written in italics because they are not English names. The manuscript must be reviewed and all names corrected. See line 2 , line 18, line 87 ,line 166, line 519, .......etc.
3-The authors used 95 isolates of Pseudomonas that were isolated during 10 years, but there is no detailed information about the isolates, the sources of isolation, and the method of isolation, and the authors did not show anything about those isolates.
4-The authors did not explain why the phage were selected from the vB_PaeM-G11 type and Figure 1 does not provide a scientific explanation for this selection.
5-The authors did not mention which statistical method was used in analyzing the results, although there are standard deviation bars in Figure 1. Some of the results require a statistical analysis to identify the significant differences.
6-Conclusions The manuscript contains some results that need to be raised and modified.
Author Response
Response to Reviewer 3 Comments
Point 1: The abstract of the manuscript is very weak and needs to be supported by adding some of the obtained results.
Response 1: We really appreciate your comments and suggestions. In the revised manuscript, we have made some additions to the abstract section.
Point 2: The scientific names of living organisms are written in italics because they are not English names. The manuscript must be reviewed and all names corrected. See line 2 , line 18, line 87, line 166, line 519, .......etc.
Response 2: All phage names have been italicized in the revised manuscript.
Point 3: The authors used 95 isolates of Pseudomonas that were isolated during 10 years, but there is no detailed information about the isolates, the sources of isolation, and the method of isolation, and the authors did not show anything about those isolates.
Response 3: Detailed information on the 95 Pseudomonas strains isolated from the Antarctic and Arctic is included in Table S3 of the Supplementary Material, which contains the isolation Country of all strains, the source of the isolated samples, and the high similarity type strains of pseudomonas matched to the 16S rDNA gene sequence. All strains were obtained in pure culture by picking single colonies from dilution and spread plates and conserved at China Center for Type Culture Collection (CCTCC), as shown in Lines 429-432.
Point 4: The authors did not explain why the phage were selected from the vB_PaeM-G11 type and Figure 1 does not provide a scientific explanation for this selection.
Response 4: Based on the phylogenetic characteristics and genomic analysis of vB_PaeM-G11, we believe that vB_PaeM-G11 phage represents a new genus in the Autographiviridae family. The tail of this bacteriophage is too small to be clearly visible in electron microscopy photos (Figure 1).
Point 5: The authors did not mention which statistical method was used in analyzing the results, although there are standard deviation bars in Figure 1. Some of the results require a statistical analysis to identify the significant differences.
Response 5: Line 151, Line 155, Lines 473-476 – Thank you for your suggestion, which we have added to the manuscript. In Figure 1, the data shown are average values from triplicate experiments, and error bars indicate standard deviation (SD). Data analysis and presentation of results were achieved by GraphPad 8, and one-way analysis of variance (ANOVA) was used to identify the significant differences.
Point 6: Conclusions The manuscript contains some results that need to be raised and modified.
Response 6: Lines 583-598 – In the revised manuscript, we have made some additions and modifications to the conclusion.
Round 2
Reviewer 2 Report
The authors have made changes to the manuscript, so I consider it can be accepted for publication.
Reviewer 3 Report
Dear Editors,
The authors made all the required adjustments to the manuscript to make it more effectively, and I now recommend that it be published as is.